



# Distribution of hydrogen peroxide over Europe during the BLUESKY aircraft campaign

Zaneta Hamryszczak[1], Andrea Pozzer[1], Florian Obersteiner[2], Birger Bohn[3], Benedikt Steil[1], Jos Lelieveld[1,4] and Horst Fischer[1]

[1]Atmospheric Chemistry Department, Max Planck Institute for Chemistry, Mainz, 55128, Germany
[2]Karlsruhe Institute of Technology, Karlsruhe, 76131, Germany
[3]Institute of Energy and Climate Research, IEK-8: Troposphere, Forschungszentrum Jülich GmbH, Jülich, 52428, Germany
[4]Climate and Atmosphere Research Center, The Cyprus Institute, Nicosia, 1645, Cyprus

*Correspondence:* Zaneta Hamryszczak (z.hamryszczak@mpic.de) and Horst Fischer (horst.fischer@mpic.de)

**Abstract.** In this work we present airborne in situ trace gas observations of hydrogen peroxide ($H_2O_2$), and methyl hydroperoxide (MHP) estimated from measurements of the sum of organic hydroperoxides over Europe during the Chemistry of the Atmosphere – Field Experiments in Europe (CAFE-EU, also known as BLUESKY) aircraft campaign. The campaign took place in May/June 2020 over Central and Southern Europe with two additional flights dedicated to the North Atlantic Flight Corridor. Airborne measurements were performed on the High Altitude and LOng-range (HALO) research operating

out of Oberpfaffenhofen (Germany). We report average mixing ratios for $H_2O_2$ of $0.32 \pm 0.25$ ppb$_v$, $0.39 \pm 0.23$ ppb$_v$ and $0.38 \pm 0.21$ ppb$_v$ within the upper and middle troposphere and the boundary layer over Europe, respectively. Vertical profiles of measured $H_2O_2$ reveal a significant decrease in particular above the boundary layer, compared to previous observations, most likely due to cloud scavenging and subsequent rainout of soluble species. In general, the expected inverted c-shaped vertical trend with maximum hydrogen peroxide mixing ratios at $3 - 7$ km was not found during BLUESKY. This contrasts

with observations during previous air-borne studies over Europe, i.e., $1.64 \pm 0.83$ ppb$_v$ during the HOOVER campaign and $1.67 \pm 0.97$ ppb$_v$ during UTOPIHAN-ACT II/III. Simulations with the global chemistry-transport model EMAC partly reproduce the strong effect of rainout loss on the vertical profile of $H_2O_2$. A sensitivity study without $H_2O_2$ scavenging performed using EMAC confirms the strong influence of clouds and precipitation scavenging on hydrogen peroxide concentrations. Differences between model simulations and observations are most likely due to difficulties in the simulation

of wet scavenging processes due to the limited model resolution.

## 1 Introduction

Hydrogen peroxide ($H_2O_2$) and related organic hydroperoxide (ROOH) species have been investigated as atmospheric trace gases for many decades, and in the 1970s hydrogen peroxide was identified as a key agent in the acidification of clouds and

rain through its oxidation of sulfur dioxide into sulfuric acid (Hoffmann and Edwards, 1975; Penkett et al., 1979; Robbin



Martin and Damschen, 1981; Kunen et al., 1983; McArdle and Hoffmann, 1983; Calvert et al., 1985). Related studies confirmed the role of $H_2O_2$ as an oxidizing agent in clouds where it accelerates the conversion of $NO_2$ to $HNO_3$ (Damschen and Martin, 1983; Lee and Lind, 1986). Efforts have also been made to characterize and analyze the amount and the chemical pathways of hydrogen peroxide in clouds (Kelly et al., 1985; Olszyna et al., 1988; Sakugawa et al., 1990; Sakugawa et al., 1993). Furthermore, gas-phase hydroperoxides are a reservoir for peroxy radicals ($HO_x$), which are well known for their contribution to the self-cleaning properties of the atmosphere (Levy, 1971; Lelieveld and Crutzen, 1990; Crutzen et al., 1999). The main source of gaseous hydrogen peroxide is the self-reaction of $HO_2$ radicals derived from the oxidation of carbon monoxide (CO) and other trace gases by OH radicals, which are formed during photolysis of ozone and the subsequent reaction of the formed $O^1D$ with water vapor (Crutzen, 1973; Logan et al., 1981; Kleinman, 1986; Lightfoot et al., 1992; Reeves and Penkett, 2003). The formation of the most prominent organic hydroperoxide, methyl hydroperoxide (MHP), results from the reaction of $HO_2$ with methyl peroxy radical ($CH_3OO$) derived from methane oxidation by OH (Levy, 1971). The formation of gaseous hydroperoxides strongly depends on the chemical composition of the troposphere as well as on meteorological conditions. Thus, mixing ratios of $H_2O_2$ and $CH_3OOH$ are primarily controlled by the mixing ratios of $O_3$, $H_2O$, CO, $CH_4$ and $NO_x$ ($NO + NO_2 = NO_x$) as well as by UV radiation. Hydroperoxide levels depend to a large extent on available peroxy radicals and therefore on $O_3$ and $NO_x$ species as the key promotors and suppressors of hydroperoxide synthesis, since peroxy radicals generally react faster with NO than they recombine. Consequently, the budget of available peroxides is influenced by the levels of ambient $NO_x$ (Campbell et al., 1979; Jaeglé et al., 1999; Lee et al., 2000). The amount of available $H_2O_2$ in the troposphere depends further on the presence of water vapor. With increasing altitudes and latitudes, the concentration of water vapor becomes the most prominent limiting factor for precursor production. With increasing altitude, the concentration of water vapor decreases, while photolytic activity simultaneously increases, and the role of hydroperoxides as a source of $HO_x$ becomes more prominent, leading to a decrease in hydrogen peroxide (Heikes et al., 1996b; Jaeglé et al., 1997; Faloona et al., 2000; Jaeglé et al., 2000). With increasing latitude, the zenith angle decreases resulting in reduced UV radiation, while the amount of water is also reduced. Therefore, the availability of hydroperoxide precursors and consequently of hydroperoxides decreases towards the poles (Jacob and Klockow, 1992; Perros, 1993; Slemr and Tremmel, 1994; Snow, 2003; Snow et al., 2007). Due to the strong sensitivity of hydrogen peroxide to deposition processes, its high solubility and pronounced mixing within the boundary layer, levels of $H_2O_2$ are limited at low altitudes where dry deposition and rainout remove the species from the troposphere. Consequently, the maximum mixing ratio can be expected above the boundary layer at 2 – 5 km resulting in a characteristic inverted c-shaped vertical profile with increasing altitude (Hall and Claiborn, 1997; Hall et al., 1999). An analogous, but less pronounced vertical profile, due to lower sensitivity towards deposition processes, is expected for organic hydroperoxides (Palenik et al., 1987; Weinstein-Lloyd et al., 1998; Snow, 2003; Snow et al., 2007).

Clouds play a significant role in the budget of hydroperoxides in the atmosphere. Cloud uptake and subsequent rainout of hydroperoxides in the aqueous phase have a considerable impact on the distribution of hydrogen peroxide. $H_2O_2$ is taken up by water droplets, dissociated and partially consumed by aqueous-phase reactions within clouds (Sakugawa et al., 1990). Previous studies have reported that despite the low volume fraction of clouds in the troposphere, levels of hydroperoxides and





65 their precursors are decreased by clouds, leading to reduced oxidation processes and therefore, diminished self-cleaning efficiency of the atmosphere. Moreover, cloud-mediated upward transport processes as well as precipitation-induced downward transport of soluble trace gases and particulate matter play key roles in the vertical distribution of many species (Lelieveld and Crutzen, 1994). Additionally, scattering, reflection and diffusion of solar radiation, which takes place within, above and below clouds, leads to modification of photolysis rates. Therefore, changes of soluble species as well as influences

70 on the chemical processes and the tropospheric redistribution caused by clouds have to be considered (Madronich, 1987; Edy et al., 1996). Finally, the effective separation of soluble and insoluble gases and the consecutive perturbation of the gas-phase chemistry balance has a great impact on the budget of the species (Lelieveld and Crutzen, 1991).

The amount and fate of $H_2O_2$ in the dynamic multi-phase cloud system is determined by the distribution of its precursors as well as by the partitioning of $H_2O_2$ between gas and liquid phases. Here, the balance between $H_2O_2$ as well as $HO_x$ ($OH + HO_2$

75 $= HO_x$) in both cloud phases is determined by the Henry coefficient and the presence of other interacting species (Brimblecombe and Dawson, 1984; Warneck, 1991, 1994). Generally, gas-phase production of OH is suppressed within clouds due to a significant pH-dependent uptake of $HO_2$ into the aqueous cloud phase. Further, due to its high Henry´s law constant, a critical amount of hydrogen peroxide itself is transferred into the aqueous phase as well. On the other hand, the aqueous phase of clouds can be an efficient source of these species as a result of cloud evaporation, droplet elevation and freezing.

80 Earlier studies report mixing ratios of hydrogen peroxide in the gaseous cloud phase of between 0.1 and 0.2 $ppb_v$ and concentrations of $10^{-7} – 10^{-4}$ mol $L^{-1}$ in the aqueous phase (Zuo and Hoigné, 1993). The budget of hydrogen peroxide within clouds depends further on conditions such as solar radiation, temperature, concentrations of volatile organic compounds (VOC) and the liquid water content that impact the mixing ratio of the trace gas. Enhanced levels of $NO_x$ and $SO_2$ have a negative effect on the total hydrogen peroxide concentration (Kelly et al., 1985; Sakugawa et al., 1990). The cloud scavenging effect

85 on MHP has to be considered as well. Despite the relatively low uptake of the species and its direct precursor $CH_3OO$ by cloud droplets, the production of MHP is reduced as a result of the reduced availability of OH. Thus, overall clouds lead to a loss of MHP, but to a far lesser extent than for hydrogen peroxide. Numerous reactions within the aqueous phase of clouds have to be distinguished from comparable processes in the gas phase. Here, a variety of reaction paths depending on cloud water pH and the presence of transition metals as well as related ionic species (especially in continental clouds) derived from anthropogenic

90 and mineral sources have to be considered (Kormann et al., 1988; Zuo and Hoigne, 1992; Anastasio et al., 1994; Zuo and Deng, 1997).

The following sections (Sect. 2 and 3) introduce the BLUESKY project and give a brief description of the experimental and modelling techniques as well as the measurement framework. In Section 4 we present the results and discuss the measurements in comparison with simulated data and with former campaigns and examine hydrogen peroxide uptake and release processes

95 in clouds based on a case study over Frankfurt airport. Here we will show that although the BLUESKY campaign was performed under lockdown conditions, we find that reduced $H_2O_2$ mixing ratios in comparison to the HOOVER and UTOPIHAN-ACT campaigns as well as the EMAC simulations are not explained by chemical but rather by meteorological



conditions. This study gives a general overview on the distribution of the species in mostly clouded environments. Hence, the presented work highlights the impact of cloud scavenging and rainout processes on the budget of the species in the troposphere.

## 2 BLUESKY campaign description

The purpose of the airborne BLUESKY campaign was to investigate how reduced emissions from anthropogenic sources due to the COVID-19 pandemic and the related shutdown were impacting the chemistry and physics of the atmosphere over Europe. To this end, the campaign obtained an overview of the distribution of a large suite of trace gases and aerosols. The decrease in air pollution and aircraft emissions provided a unique opportunity for analysis of the resulting changes in the atmosphere. The reduced pollution levels gave rise to anomalous blue skies, hence the name "BLUESKY" (Voigt et al., 2021, submitted). The project was carried out in May and June 2020 covering an area from the Mediterranean region in Southern Europe (appr. 35°N) up to the North Atlantic flight corridor (appr. 60 °N). During the measurement period, eight measurement flights were carried out with the German High Altitude and LOng-range research aircraft (HALO). The entirety of the flight tracks of HALO during the campaign color-coded by flight altitude is presented in Fig.1. Flights over the North Atlantic flight corridor were not included in this study, since they were performed entirely in the lower stratosphere.



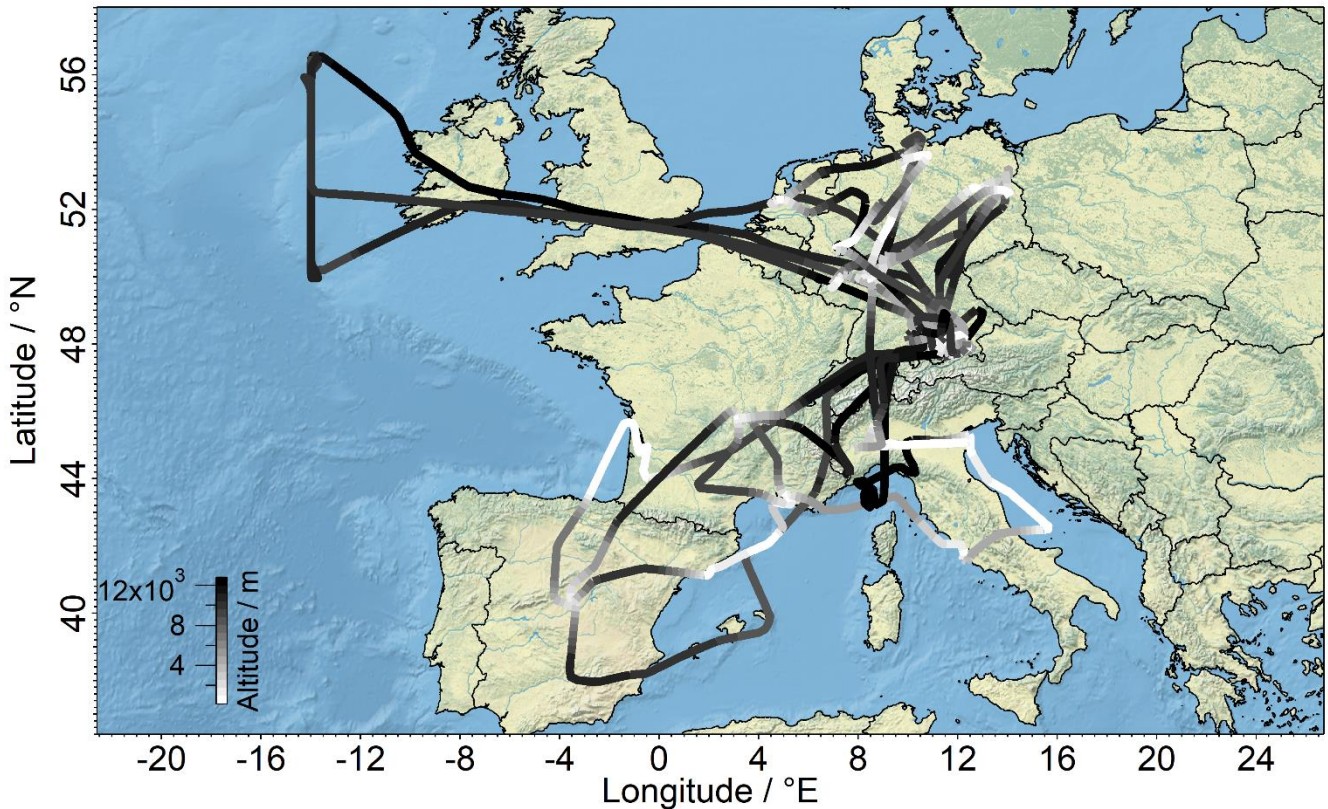

**Figure 1. Flight tracks of the BLUESKY measurement campaign over Europe color-coded by GPS altitude. All flights were performed from the flight base in Oberpfaffenhofen, Germany (48 °N, 11 °E).**

The measurement flights ranged in altitude from a few tens of meters above the earth's surface to approx. 14 km, i.e., reaching

beyond the tropopause into the lower stratosphere. Individual flights were performed between 7:00 and 17:00 UTC on eight

different days, with at least one maintenance day between flights. Vertical profiles were generally obtained during fly-byes

near main European airports and urban areas in order to sample air from emission sources from the earth's surface up to the

tropopause region. The flights were performed from the flight base of the German Aerospace Center (Deutsches Zentrum für

Luft- und Raumfahrt), DLR in Oberpfaffenhofen, Germany.

The Monthly Bulletin on the Climate in WMO Region IV Europe and Middle East for the months of May and June 2020

published by DWD (Deutscher Wetterdienst) indicate a regional monthly averaged cloud coverage span of 37.5 – 75%. The

total precipitation in May and June varied locally between 100 – 200 mm per month and 100 – 150 mm per month, respectively,

which amounts to 80 – 125% of the values relative to the reference period from 1981 until 2010. The average air temperature

at 2 m above the surface was approximately 2 °C higher than during the reference period 1981 – 2010. An overview on the

average meteorological conditions based on ERA5 reanalysis data generated by the Copernicus Climate Change Service

(Hersbach et al. 2019) is presented in the Supplement (Fig. S1). Additionally, meteorological conditions for single flight days,





which show cloud cover fractions of approximately 60% and higher with many rain events along the flight tracks at altitudes of 2 – 7 km, are also presented in the Supplement (Fig. S2).

## 3 Methods

### 3.1 Hydrogen peroxide measurement

Hydrogen peroxide and the sum of organic hydroperoxides were measured using a wet chemical monitoring system, HYdrogen Peroxide and Higher Organic Peroxide monitor (HYPHOP; Stickler et al., 2006; Klippel et al., 2011; Bozem et al., 2017; Hottmann et al., 2020) based on the work of Lazrus et al., 1985; Lazrus et al., 1986). Ambient air was probed via a 1/4" PFA tubing installed in a stainless-steel inlet setup (TGI; trace gas inlet). From the inlet, the peroxide species were sampled via a
bypass. In order to avoid any pressure and therefore, airflow inconsistencies a Constant Pressure Inlet setup (CPI) was used, which consists of a Teflon-coated membrane pump (type MD 1C; Vacuubrand, Wertheim, Germany) and a pressure control unit that adjusts the pump speed to a line pressure of 1000 hPa. The sampling efficiency of the inlet for $H_2O_2$ was determined to be 0.52. After passing through the CPI, ambient air enters the instrument and passes through a sampling coil with a buffered sampling solution (potassium hydrogen phthalate/NaOH; pH 6) with a stripping efficiency of 1 for hydrogen peroxide and 0.6
for MHP (Lee et al., 2000). The resulting peroxide solution was separated into two channels, where p-hydroxyphenyl acetic acid (POPHA) and horseradish peroxidase (HRP) were added. The stoichiometric reaction yields the chemiluminescent compound 6,6'- dihydroxy-3,3'-biphenyldiacetic acid, which is measured by means of fluorescence spectroscopy with a Cd pen ray lamp at 326 nm. The resulting hydroperoxide-specific fluorescence (Guilbault et al., 1968) at 400 – 420 nm was detected using a photomultiplier tube for each channel. In order to specifically measure hydrogen peroxide, this species was
selectively destroyed by catalase in one channel (Channel B). $H_2O_2$ can then be calculated as the difference between the sum of all hydroperoxides (Channel A) and the entirety of the remaining organic hydroperoxides (Channel B). This measurement technique does not provide mixing ratios for individual organic hydroperoxides, however. Nevertheless, previous studies indicate that methyl hydroperoxide is the most prominent free tropospheric component of organic hydroperoxides (90 – 100 %; Heikes et al., 1996a; Jackson and Hewitt, 1996; Walker et al., 2006; Hua et al., 2008). For this study we assumed that MHP is
the sole component of organic hydroperoxides that passes the inlet and is unaffected by any further losses and scaled the signal of Channel B with the sampling efficiency for MHP based on the stripping efficiency. Thus, MHP used in this paper is an upper limit for the actual MHP in the free troposphere. In particular, in the boundary layer other organic hydroperoxide species are expected to contribute to the signal in Channel B. Based on previous studies, HMHP (hydroxymethyl hydroperoxide) and to a lesser extend of PAA (peroxyacetic acid) and EHP (ethyl hydroperoxide) contribute significantly to the total amount of
organic hydroperoxide mixing ratios at low altitudes (Fels and Junkermann, 1994; Slemr and Tremmel, 1994; Valverde-Canossa et al., 2005; Hua et al., 2008).



The catalase efficiency for the destruction of $H_2O_2$ in Channel B was determined via liquid calibration of the instrument prior to the measurement at $0.95 - 0.98$. For the simultaneous liquid calibration of both channels, a $H_2O_2$ standard ($0.98$ µmol $L^{-1}$)

produced in a serial dilution from a stock solution was used. In order to estimate the sampling efficiency, a calibration gas was analyzed every second day during the field campaign. The calibration gas was created by a LDPE permeation source filled with 30% hydrogen peroxide embedded in a temperature-controlled oven at 35 °C and flushed with synthetic air at a rate of 60 standard cubic centimeters per minute (sccm). The defined amount of hydrogen peroxide gas was diluted with approximately 2300 sccm purified ambient air. The permeation gas can be calibrated by bubbling the gas through a water-

filled flask followed by photometric examination via UV spectroscopy using the $TiCl_4$ method described by Pilz and Johann, 1974). The in-flight background measurements were performed using peroxide-free air generated by a cartridge filled with hopcalite (type IAC-330) and silica gel (type IAC-502; Infiltec, Speyer, Germany).

To account for the sensitivity of hydrogen peroxide towards metal ions in the Fenton reaction (Graedel et al., 1986; Zepp et al., 1992; Weinstein-Lloyd et al., 1998) as well as towards sulfur dioxide ($SO_2$), ethylene diamine tetra acetic acid (EDTA)

and formaldehyde (HCHO) were added to the stripping solution. Further, the data was corrected for existing positive ozone interference by subtraction of $0.016$ ppb$_v$ $H_2O_2$/100 ppb$_v$ $O_3$. The interference was derived by plotting hydrogen peroxide mixing ratios vs. ozone mixing ratios in the lower stratosphere, assuming that ambient $H_2O_2$ is zero above the tropopause. Due to instrumental issues caused by hopcalite contaminations during the campaign, the uncertainty of the ozone interference was further estimated as 27% at $0.16$ ppb$_v$ hydrogen peroxide.

The total measurement uncertainty (TMU) of the instrument was calculated as:

$$TMU = \sqrt{((P)^2 + (US)^2 + (UIE)^2 + (UOI)^2)} \tag{1}$$

and was derived by considering the instrument's Precision (P), Uncertainty of the Standard and Inlet Efficiency (US; UIE) as well as the Uncertainty of the Ozone Interference (UOI). The precision was determined as 0.3% @ 5.1 ppb$_v$ for hydrogen peroxide and 0.2% @ 5.4 ppb$_v$ for organic peroxides. The uncertainty of the standard was included in instrument precision

calculations. The uncertainty of the inlet efficiency was calculated to be 5%. The calculated total measurement uncertainty was therefore determined at 28% for hydrogen peroxide and 40% for the sum of organic peroxides. Moreover, the time resolution of the instrument was determined to be 2 min based on the signal rise time from 10% to 90%. Based on the average cruise speed of the research aircraft of $179 \pm 51$ m/s the spatial resolution of the 2 min sample was estimated as 21.5 km. The detection limit, derived from 2 sigma uncertainty of 37 background measurements was $0.035$ ppb$_v$ for hydrogen peroxide and

$0.013$ ppb$_v$ for organic peroxides. For the purposes of this study, the obtained peroxide data was limited to measurements within the troposphere by removing all data points with ozone mixing ratios higher than 100 ppb$_v$.

## 3.2 Measurement of other species

The measurements of ozone were carried out with a chemiluminescence detector calibrated by a UV photometer of the Fast AIrborne Ozone instrument, FAIRO (Zahn et al., 2012). Upward and downward spectral actinic flux density was recorded





with two spectroradiometers (Bohn and Lohse, 2017). Water vapor mixing ratios as well as humidity measurements were obtained with the Sophisticated Hygrometer for Atmospheric ResearCh (SHARC) based on a tunable diode laser (TDL) setup (Krautstrunk and Giez, 2012). GPS data as well as temperature, pressure, wind speed and true air speed were obtained using the BAsic HALO Measurement And Sensor System, BAHAMAS. The list of campaign instrumentation as well as the complimentary measurement method, TMU, and references regarding the use of each technique are given in Table 1.


**Table 1. Overview of observed species with corresponding measurement method, total measurement uncertainty (TMU) and references regarding the instrumentation.**

| Measurement | Method | TMU | References |
|---|---|---|---|
| $O_3$ | Chemiluminescence + UV absorption | 2.5% | Zahn et al., 2012 |
| Actinic flux density | Spectroradiometer | 7 – 8 % (15% for $j(H_2O_2)$) | Bohn and Lohse, 2017 |
| $H_2O$ | TDLAS | 5% | Krautstrunk and Giez, 2012 |

### 3.3 ECHAM/MESSy Atmospheric Chemistry (EMAC) model

In this study, we used the global numerical 3-D model EMAC (ECHAM/MESSy for Atmospheric Chemistry, Jöckel et al., 2010), which simulates numerically the chemistry and dynamics of the troposphere and the stratosphere. EMAC incorporates a variety of submodels addressing chemical and metrological processes and their interactions with marine, continental and anthropogenic environments (Jöckel et al., 2006). The basis atmospheric model is the 5[th] generation of the European Centre HAMburg general circulation model (ECHAM5, Roeckner et al., 2003; Roeckner et al., 2006). For standardized data exchange

between submodels and the base model, the Modular Earth Submodel System (MESSy; Jöckel et al., 2005; Jöckel et al., 2006; Jöckel et al., 2016) was used. The Module for Efficiently Calculating the Chemistry of the Atmosphere (MECCA) submodel was used to simulate stratospheric and tropospheric gaseous and heterogeneous chemistry (Sander et al., 2005; Sander et al., 2011; Sander et al., 2019). For the simulation of aqueous phase chemistry in clouds and wet scavenging processes the Scavenging of Tracers (SCAV; Tost et al., 2006) submodel was applied. Primary emissions as well as dry deposition of

atmospheric trace gases and aerosols were simulated by submodels ONLEM, OFFLEM, TNUDGE and DRYDEP (Kerkweg et al., 2006a; Kerkweg et al., 2006b). The simulations of anthropogenic emissions were based on CAMS-GLOB-ANTv4.2 (Granier et al., 2019), which uses emission data provided by the EDGARv4.3.3 inventory developed by the European Joint Center (JRC; Crippa et al., 2018) and CEDS emissions (Hoesly et al., 2018). Emission reduction coefficients were additionally adapted to lockdown conditions in Europe based on the work of Guevara et al. (2021). A detailed description of the emission

submodels as well as their modifications are presented in the work of Reifenberg et al. (2021). The horizontal resolution of the model in this study is T63 (i.e., roughly 1.8°×1.8°) and the vertical resolution consists of 47 levels up to 0.01 hPa. The



simulated data has a time resolution of 5 min. Importantly, for the purpose of comparison with the observations, the model results were interpolated along the GPS flight tracks with the S4D submodel (Jöckel et al., 2010).

## 4 Results

**4.1 Distribution of hydrogen peroxide and comparison with previous observations over Europe**

During the previous field campaigns UTOPIHAN-ACT (Upper Tropospheric Ozone: Processes Involving HOx And NOx: The Impact of Aviation and Convectively Transported Pollutants in the Tropopause Region) and HOOVER (HOx OVer EuRope), numerous measurement flights were performed in 2002 – 2004 and 2006 – 2007 over Europe (Colomb et al., 2006; Stickler et al., 2006; Klippel et al., 2011). The flight tracks during the two campaigns covered a similar latitudinal and
altitudinal range. Thus, parts of both campaigns performed during spring and summer seasons within the latitudinal range 40 – 55 °N can be compared with the outcomes of our measurements. The comparison described below is restricted to HOOVER II (July 2007) and UTOPIHAN-ACT II (March 2003) and III (July 2003) to ensure overlap with the late spring/early summer measurements presented here. The latitudinal distribution of hydrogen peroxide during the three campaigns is presented in Fig. 2 as a function of latitude for three altitude ranges within the troposphere (boundary layer (BL) from $0 < 2$
km, middle troposphere (MT) from $2 < 6$ km, and upper troposphere (UT) from $6 – 14$ km). The presented mean values of the datasets with 2 min resolution are binned into subsets of 2.5° of latitude for the entirety of each tropospheric layer. The datasets can be further studied by comparing the vertical profiles of all campaigns, as displayed in Fig. 3a. The mean values of the data are binned into subsets of 0.5 km of altitude. The medians and means ($\pm$ 1 sigma) of hydrogen peroxide mixing ratios calculated with 2 min resolution within the range $37.5 – 52.5$ °N for each campaign are listed in Table S1 (Supplement).



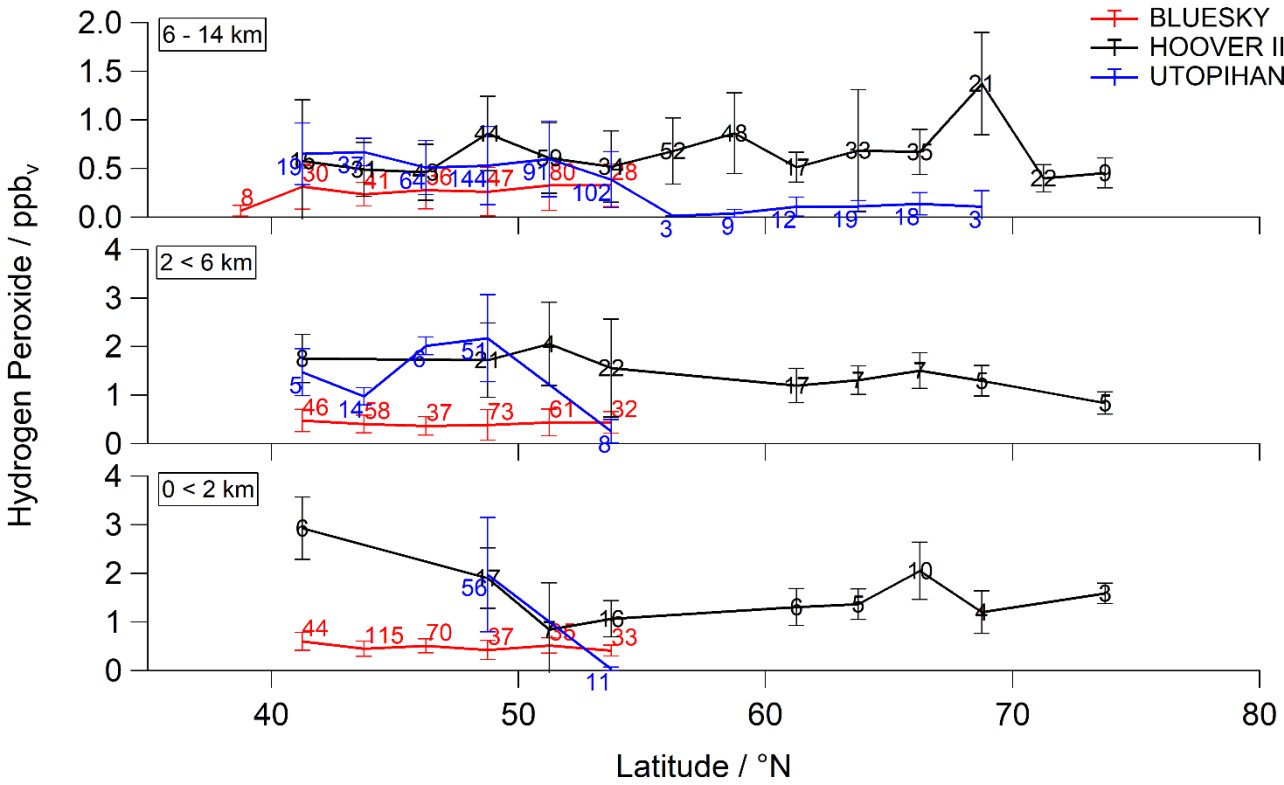

**Figure 2.** Latitudinal dependence of hydrogen peroxide concentrations (mean ± 1 sigma) compared to former campaigns (red: BLUESKY; black: HOOVER II; blue: UTOPIHAN-ACT). The data with 2 min time resolution was subdivided into three atmospheric layers, upper troposphere, middle troposphere and boundary layer (from top to bottom) with mean values binned for 2.5° of latitude for each tropospheric layer. The corresponding numbers indicate the total amount of data points per bin.





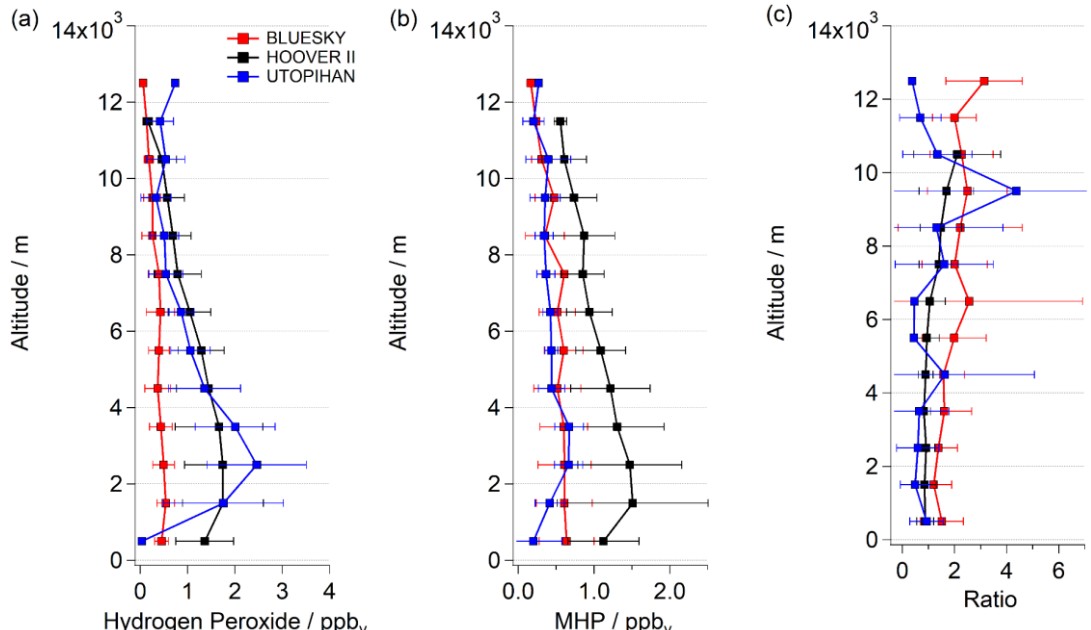

**Figure 3. Comparison of vertical hydrogen peroxide profiles (a), MHP (b) and MHP/H₂O₂ ratio (c) during BLUESKY (red) with outcomes of the earlier campaigns, HOOVER II (black) and UTOPIHAN-ACT II/III (blue). The data was plotted as mean ± 1 sigma.**

The observed distribution within the UT and the latitudinal range 37.5 – 52.5 °N amounts to a mean (median) mixing ratio of $0.28 \pm 0.22$ (0.24) $ppb_v$ for BLUESKY, which is lower in comparison to the previously measured $0.67 \pm 0.43$ (0.56) $ppb_v$ during HOOVER II and $0.47 \pm 0.36$ (0.47) $ppb_v$ during UTOPIHAN-ACT. In both lower tropospheric layers (0 – 6 km), the intercomparison is dominated by the high variability of the mixing ratios. The mean mixing ratios for hydrogen peroxide show further pronounced reductions with values up to 72% and 76% lower compared to HOOVER and UTOPIHAN-ACT for both

lower tropospheric layers. Mixing ratios of $0.42 \pm 0.25$ (0.37) $ppb_v$ within MT and $0.48 \pm 0.17$ (0.48) $ppb_v$ in BL were determined. During previous campaigns much higher hydrogen peroxide mixing ratios of $1.49 \pm 0.71$ (1.33) $ppb_v$ during HOOVER and $1.74 \pm 0.97$ (1.80) $ppb_v$ during UTOPIHAN were measured at altitudes of 2 – 6 km. Also, the results within the boundary layer display a significant discrepancy with the previously measured $1.59 \pm 0.78$ (1.48) $ppb_v$ for HOOVER and $1.65 \pm 0.16$ (1.32) $ppb_v$ during UTOPIHAN. The observed mixing ratios are only in approximately 30% agreement with

previous results.

A significant tendency towards lower mixing ratio for hydrogen peroxide during the BLUESKY project was observed, in particular at altitudes below approx. 7 km (Fig. 3a). The altitude profile does not agree very well with the described inverted c-shaped distribution trend in the literature (Klippel et al., 2011; Bozem et al., 2017). The most striking feature of the BLUESKY observations is the absence of a local maximum of H₂O₂ above the BL. Instead, the mixing ratio is rather constant

in the 3 – 7 km range. We hypothesize that these differences between the campaigns predominantly originate from differences





in the meteorological rather than chemical conditions. We will show that the low observed mixing ratios of hydrogen peroxide during BLUESKY are most likely caused by enhanced wet scavenging processes due to a pronounced presence of clouds at altitudes of 3 – 7 km. An analysis of hourly cloud coverage at altitudes of 2 – 7 km based on ERA5 reanalysis generated using Copernicus Climate Change Service information (Hersbach Hersbach, H., Bell, B., Berrisford, P., Biavati, G., Horányi, A.,

Muñoz Sabater, J., Nicolas, J., Peubey, C., Radu, R., Rozum, I., Schepers, D., Simmons, A., Soci, C., Dee, D., Thépaut, J-N., 2018; Fig. 2S) for single flights (BLUESKY) shows high average values. Further, based on log book information of all campaigns, there was a pronounced presence of clouds during BLUESKY in comparison with previous airborne measurements. We have observed a high number of cloud and rain events along the flight tracks during the BLUESKY campaign compared to the mostly cloud-free measurement conditions during HOOVER and UTOPIHAN-ACT.

Generally, the presence of clouds has a marked impact on $H_2O_2$ but a much smaller effect on MHP. MHP is less sensitive to wet deposition due to smaller Henry's coefficient ($2.2 \cdot 10^4$ mol $L^{-1} \cdot atm^{-1}$ at 298 K for MHP in contrast to $7.4 \cdot 10^4$ mol $L^{-1} \cdot atm^{-1}$ at 298 K for $H_2O_2$). Therefore, the concentration ratio of both species can be an indicator of cloud presence (Heikes et al., 1996b; O'Sullivan et al., 1999; Snow, 2003; Snow et al., 2007; Klippel et al., 2011). The assumption of cloud processing via ratio comparison is derived from the fact that highly soluble species are transferred into the aqueous

phase of clouds, where they are removed by reactions with other soluble species or by precipitation (Crutzen and Lawrence, 2000). Consequently, an increase in the ratio between MHP and hydrogen peroxide of ≥ 1 can ensue as a result of deposition processes within clouds. Please note that due to the characteristics of the measurement technique, which derives the estimated MHP mixing ratio as its tropospheric upper limit (Sect. 3.1.), the vertical trend of MHP and therefore also the MHP/$H_2O_2$-ratio are expressed as qualitative comparisons. Vertical profiles of MHP measured during the BLUESKY and UTOPIHAN-ACT

projects are comparable, while HOOVER II displays clearly higher values (Fig. 3b). However, the vertical trends of peroxides during HOOVER II can be assumed to be about equal, leading to an MHP vs. $H_2O_2$ ratio of approximately 1. Thus, the two previous campaigns over Europe show corresponding trends with decreasing MHP/$H_2O_2$-ratios above the boundary layer, where $H_2O_2$ mixing ratios are at their maxima (Fig. 3c). In contrast, increasing ratios of MHP vs hydrogen peroxide at altitudes of 3 – 7 km were observed during the BLUESKY campaign. These increases during the BLUESKY campaign can be attributed

to the lower mixing ratio of $H_2O_2$ and are indicative of more pronounced cloud scavenging.

## 4.2 Comparison with the EMAC model

In order to test the hypothesis that hydrogen peroxide is depleted at altitudes of 3 – 7 km due to cloud scavenging, a comparison of in situ data with the output of the EMAC model was performed. The analysis of the EMAC and in situ results for different latitudinal distribution subdivided into three main tropospheric air layers is presented in the Supplement (Fig. 3S). The

comparison between the model results and observations shows a generally good agreement for the UT as well as for the majority of data in the BL. Discrepancies here at low latitudes can be seen for Mediterranean areas influenced strongly by marine air masses (Barcelona, 41° 24′ N and Rom, 41° 53′ N), where the model tends to overestimate the mixing ratio of




hydrogen peroxide by up to a factor of 3. This is most likely related to model resolution (~ 180 km x 180 km), which makes it difficult to differentiate marine from continental airmasses in coastal areas. The difference between observed and modelled

data for both tropospheric regions is not significant and a good agreement between the simulated and measured sources and sinks within the top and bottom tropospheric layers can be assumed (latitudinal distribution at 0 – 2 km and 6 – 12 km; Fig. 3S). In contrast, the model tends to overestimate $H_2O_2$ concentrations in the 2 – 6 km range. As stated above, the observed mixing ratios are generally low at 3 – 7 km, most likely related to the high impact of deposition processes within clouds. An analysis of the impact of emissions from anthropogenic sources on modelled $H_2O_2$ data shows an average difference of

approximately 2.5% between mixing ratios with and without the lockdown emissions reductions within lower tropospheric layers (1 – 7 km; Fig. 4S). Thus, emission reduction is not considered to be responsible for the strong deviation between the simulated and observed hydrogen peroxide mixing ratios (Reifenberg et al. 2021).

In order to investigate potential causes for the observed difference between observations and model simulations, we calculate the hydrogen peroxide budget based on photostationary steady-state conditions using model-simulated radical and photolysis

rate data. In the free troposphere, the production rate P of hydrogen peroxide can be calculated from Eq. (2) and the photochemical loss rate L due to photolysis and reaction with OH from Eq. (3).

$$P(H_2O_2) = k_{HO_2+HO_2} \cdot [HO_2]^2 \tag{2}$$

$$L(H_2O_2) = \left(k_{H_2O_2+OH} \cdot [OH] + j(H_2O_2)\right) \cdot [H_2O_2] \tag{3}$$

Neglecting deposition and transport processes impacting the hydrogen peroxide budget, the maximum concentration of $H_2O_2$

can be calculated as presented in Eq. (4).

$$[H_2O_2]^{PSS} = \frac{[HO_2]^2 \cdot k_{HO_2+HO_2}}{[OH] \cdot k_{H_2O_2+OH} + j(H_2O_2)} \tag{4}$$

Vertical profiles of observed, simulated and calculated $H_2O_2$ under the assumption of photostationary state conditions are displayed in Fig. 4a. Additionally, a model sensitivity study in which the scavenging of $H_2O_2$ in clouds was omitted has been incorporated.



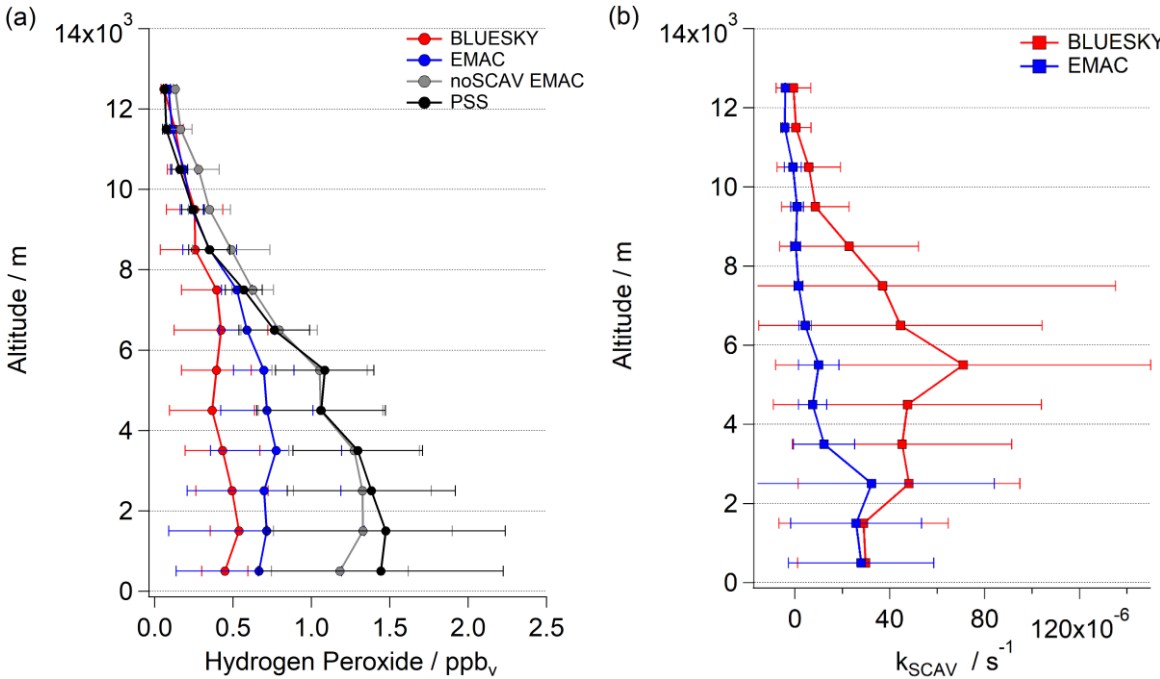

**Figure 4. Vertical profiles of observed (red), simulated (blue), reduced simulated (gray) and calculated under assumption of PSS (Photostationary Steady State) conditions (black) hydrogen peroxide (a) and calculated loss rate constant $k_{SCAV}$ by scavenging (b); red lines show observed values, while blue lines show modelled $H_2O_2$.**

The comparison of all datasets shows that no significant hydrogen peroxide loss occurs within the cloud-free layers of the
upper troposphere (above 10 km), where the resulting mixing ratios show a similar vertical trend (Fig. 4a). Both the sensitivity
study and the PSS calculation indicate further that wet scavenging in clouds followed by rainout as the ultimate removal
process forms a substantial sink for $H_2O_2$, in particular in the middle and lower troposphere. Please note that this sink is not
always associated with in-cloud conditions along the flight paths. Due to the photochemical lifetime of $H_2O_2$, which is on the
order of several days, local $H_2O_2$ mixing ratios will also depend on up-wind cloud processing (Cohan et al., 1999; Hua et al.,
2008). Although EMAC reproduces this cloud processing, the absolute mixing ratios are still overestimated, indicating a
potential underestimation of the deposition rate in the model.

Based on $[H_2O_2]^{PSS}$ the deposition loss rate constant was calculated by comparing to $[H_2O_2]^{Obs}$ and $[H_2O_2]^{EMAC}$:

$$\{k_{H_2O_2+OH} \cdot [OH] + j(H_2O_2) + k_{SCAV}\} \cdot [H_2O_2] \ = \ P(H_2O_2) \tag{5}$$

The total modelled loss rate constant due to deposition $k_{SCAV}$ based on PSS conditions (Eq. (2) and (3)) shows an
underestimation of 2.2 compared to the observationally derived constant. The vertical profile of the simulated deposition rates
is in a good agreement with the observations at low altitudes (below 2 km), where dry deposition plays a key role in the removal
of $H_2O_2$ species (Fig. 4b). At high altitudes (UT) EMAC loss rate results agree well with the observations. The majority of the





loss processes takes place within the MT (2 – 8 km). Here, EMAC underestimates the deposition impact most prominently, which is corresponding to the discrepancies between observed and modelled hydroperoxide mixing ratios (Fig. 4a).

Since the loss of hydrogen peroxide in the atmosphere strongly depends on the presence of clouds, the temporal uptake and loss of the species within cloud droplets and the permanent loss by rainout, it is important that the model correctly reproduces cloud coverage, liquid water content (LWC) and precipitation rates. In Fig. 5a, a histogram of the average total cloud coverage over all measurement days based on EMAC and ERA5 (containing modified Copernicus Climate Change Service information; Hersbach et al. 2018), respectively, indicates an underestimation of cloud coverage by EMAC in comparison with ERA5. The

discrepancy is most pronounced over Central Europe (47 – 55 °N; 6 – 15 °E) and is smaller over the North Atlantic (approximately 30 – 40 °N; - 50 – - 30 °E; Fig. S5). The comparison of the average liquid water path (LWP) based on LWC of the measurement days shows a difference of approximately 2% by EMAC in comparison to ERA5, which indicates a minor deficit in the simulated species uptake (Fig. S6). The main difference between EMAC and ERA5 arises from the comparison of the total precipitation. As shown in Fig. 5b, the model underestimates heavy rain events (> 0.5 mm s$^{-1}$) in comparison with

the ERA5 reanalysis model (modified Copernicus Climate Change Service information; Hersbach et al. 2018). With respect to ERA5, a difference by a factor of 2.2 was estimated for the entirety of the region compared to EMAC, which agrees well with the calculated ratio of the loss rates. Further, a less pronounced impact of scavenging on the hydrogen peroxide budget, and therefore higher mixing ratios, are simulated by the model.


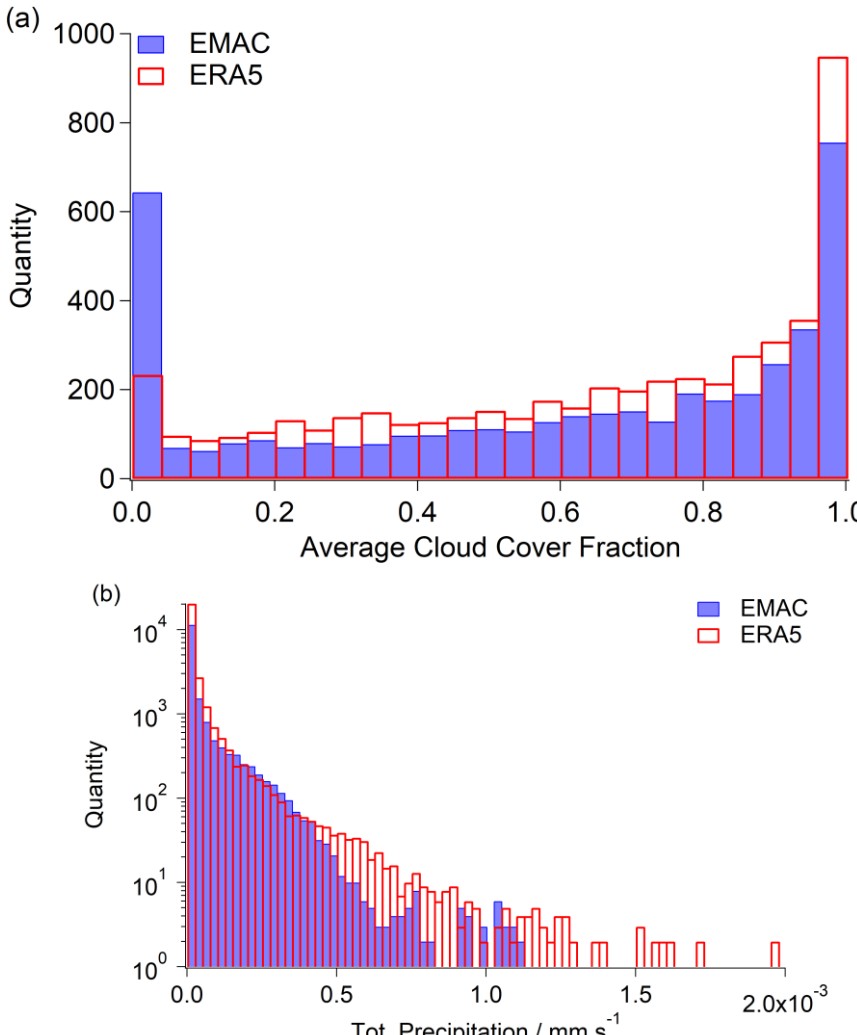

**Figure 5. Histograms of average cloud cover fraction (a) and total precipitation (b) over the North Atlantic and Europe (73 - 28°N, - 50 – 15°E) based on ERA5 (red; modified Copernicus Climate Change Service information; (Hersbach, H., Bell, B., Berrisford, P., Biavati, G., Horányi, A., Muñoz Sabater, J., Nicolas, J., Peubey, C., Radu, R., Rozum, I., Schepers, D., Simmons, A., Soci, C., Dee, D., Thépaut, J-N., 2018) and EMAC simulation (blue).**

As shown based on the difference in total precipitation between EMAC and ERA5, EMAC appears to underestimate the majority of the rainout events at the location of the flight tracks (Fig. 6). Detailed comparison of the average total precipitation between EMAC and ERA5 during the campaign can be found in the Supplement of this work (Fig. S7).

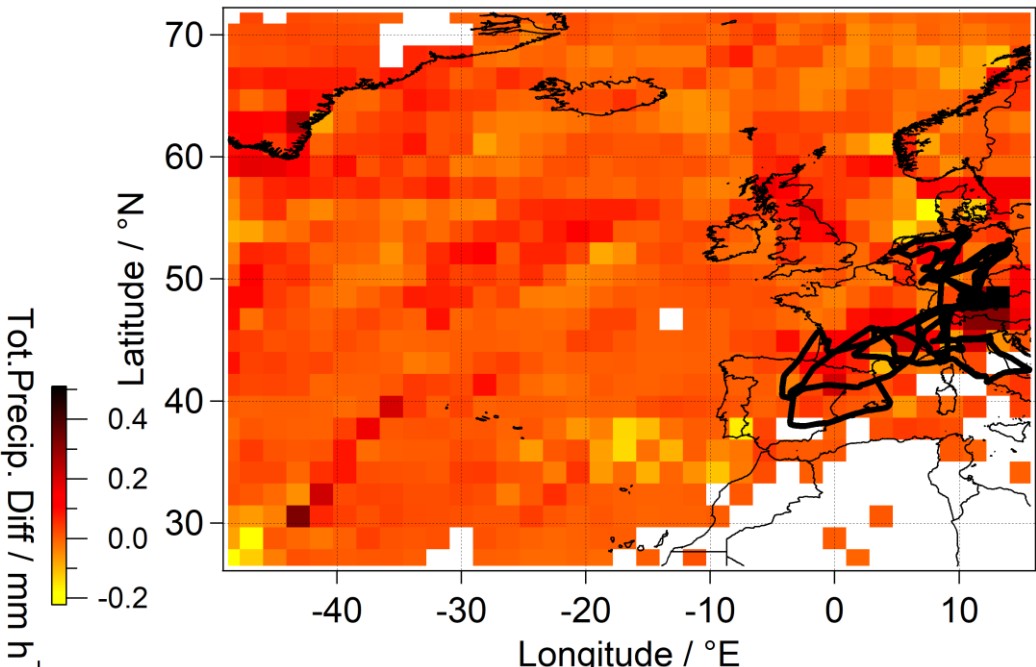

**Figure 6. Average total precipitation difference between ERA5 and EMAC over Europe and North Atlantic (73 – 28 °N,**
**-50 – 15°E; contains modified Copernicus Climate Change Service information; Hersbach et al. 2018). The North**
**Atlantic region was included in view of the $H_2O_2$ lifetime in the atmosphere and the air mass origins based on backward**
**trajectories. The performed flights are indicated in black.**

Assuming a linear dependence between ultimate removal of $H_2O_2$ in cloud droplets and precipitation as given by EMAC
simulations, total large-scale scavenging was estimated based on precipitation given by ERA5. The prediction is based on
extrapolating the linear relationship between EMAC scavenging and simulated total precipitation. The simulated scavenging
by EMAC falls short by $7.6 \cdot 10^{13}$ molecules $m^{-2} s^{-1}$ relative to the scavenging prediction based on ERA5 output (Fig. S8). The
difference between hydrogen peroxide mixing ratios obtained using EMAC and the observations (Fig. 4a) estimated as the
integral between the observed and simulated mixing ratios over the entire troposphere for the total measurement time in
molecules $m^{-2} s^{-1}$ are comparable ($7.3 \cdot 10^{13}$ molecules $m^{-2} s^{-1}$). This indicates that the underestimation of the rain rate by EMAC
relative to ERA5 is responsible for the overestimation of $H_2O_2$ in the model. As indicated by the rainout discrepancy between
EMAC and ERA5, a higher variability in scavenging can be expected along the flight tracks (Fig. 6).

Other major causes leading to the observed discrepancy might be an overestimation of peroxide sources as well as an
underestimation of its photochemical sinks. The analysis of the photolysis frequencies for both datasets showed a discrepancy
by a factor of approximately 1.5. However, the underestimation of the photolysis frequencies by the model can be partly
explained by the use of different absorption cross sections of $H_2O_2$ (Hottmann et al., 2020). Calculations of hydrogen peroxide
mixing ratios under PSS conditions based on simulated and observed photolysis frequencies are in good agreement (Fig. S9).





Therefore, the impact of photochemical loss processes on the hydrogen peroxide budget is considered as minor. An impact of peroxide precursor discrepancies cannot be determined due to the lack of $HO_x$ measurement, although the study of Reifenbach et al. demonstrates good agreement between observations and model results for those species affecting $H_2O_2$, i.e., $NO_x$, $O_3$ and

$H_2O$. Thus, the overestimation of hydrogen peroxide in the model is most likely due to underestimation of scavenging processes.

### 4.3 The fate of hydrogen peroxide below clouds

The distribution of hydrogen peroxide above, in and below clouds at Frankfurt Airport (50° 1′ 59″ N and 8° 34′ 14″ O) were measured during the BLUESKY-flight #1 and showed untypical increases in hydrogen peroxide mixing ratios at low altitudes.

The descent and ascent into and out of Frankfurt took place between 9:00 and 11:00 UTC. Fig. 7 displays the time series of the approach to Frankfurt. Mixing ratios of $H_2O_2$ from observations and EMAC are shown.

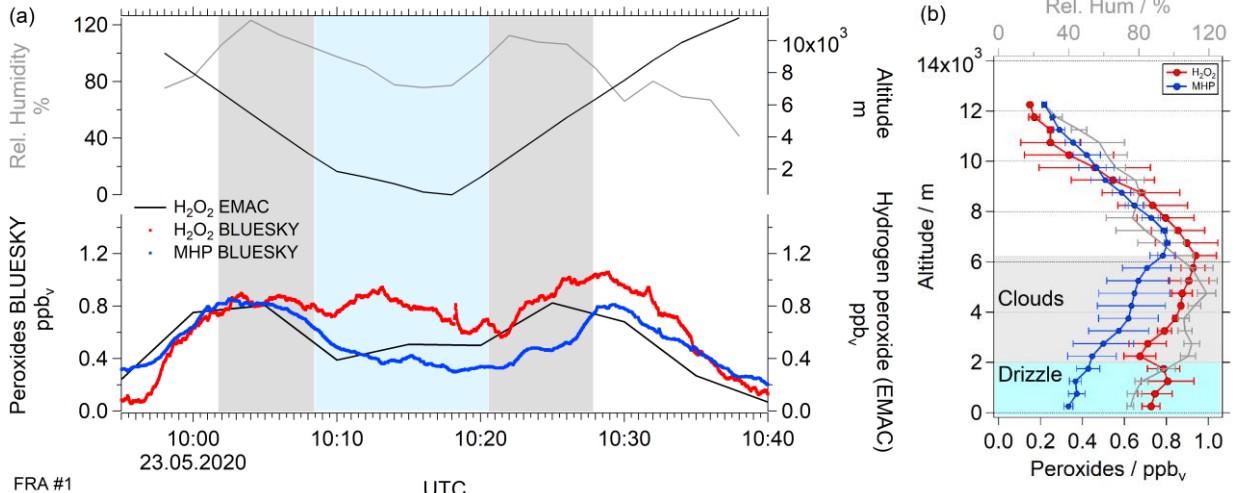

**Figure 7. Temporal series of BLUESKY flight #1 over Frankfurt (50° 1′ 59″ N and 8° 34′ 14″ O; left) and vertical**
**distribution (right) of hydrogen peroxide impacted by cloud and rain scavenging. Data were plotted for observed (red) and simulated (black) hydrogen peroxide mixing ratios, and the observed MHP mixing ratio (blue) in relation to altitude (top black) and relative humidity (light gray). Cloud scavenging and precipitation are highlighted in gray and light blue boxes, respectively. Please note, that the displayed peroxide data has a temporal resolution of 1 sec in contrast to the model resolution of 5 min.**

A relative humidity (RH) of 100% (grey areas in Fig. 7) indicates the presence of clouds. Rain was mainly observed below the clouds at low altitudes (light blue areas) at slightly lower RH. ERA 5 reanalysis (Fig. S11a) confirmed the presence of clouds at altitudes of 2 – 6 km (Flight #1). The average mixing ratio of $H_2O_2$ over Frankfurt (Fig. 7) was $0.646 \pm 0.229$ $ppb_v$. Above




the tops of the clouds, no significant increase in mixing ratios could be observed. An increase in observed hydrogen peroxide mixing ratios occurred after exiting the clouds during the descending part of the vertical profile track. Here, the $H_2O_2$

concentration exceeded 0.7 ppb$_v$ and dropped again within a short time (10 min) to 0.567 ppb$_v$. The maximum measured hydrogen peroxide mixing ratio was 0.8 ppb$_v$. The observed hydrogen peroxide mixing ratio peak might be caused by cloud scavenging which is strongest at the bottom, where the liquid water content is also highest. However, this assumption is not supported by the simultaneous MHP mixing ratio observations (Fig 7b). In contrast to hydrogen peroxide, MHP mixing ratios display a decreasing trend below 2 km. It seems that the increase in hydrogen peroxide concentrations was caused by an

additional source of this species below clouds. An analogous phenomenon was observed for measurements taken over Bordeaux (Fig. S10 and S11b).

Previous studies on the possibility of mass transfer of $H_2O_2$ from rain water to the surrounding air indicate a possible release of hydrogen peroxide to the atmosphere (Hua et al., 2008; Huang and Chen, 2010; Xuan et al., 2020). Raindrops are affected by the temperature gradient between the earth's surface and the cloud base. The negative dependence of hydrogen peroxide

solubility on temperature derived from the Henry's law constant means that an impact on the aqueous-gas phase equilibrium can be assumed. Moreover, the mass transfer coefficient is dependent on the surface-to-volume ratio of the rain drops and is diminished for large rain drops due to a smaller contact surface between the liquid and gas phase (Xuan et al. 2020). The size of the raindrops can be derived from the rain intensity, as shown by Kumar, 1985). During the vertical profile flights a light rain (drizzle) was reported, consistent with rain sum measurements (approximately $10^{-4}$ mm h$^{-1}$) and ERA5 reanalysis plots

provided by Copernicus Climate Change Service (Hersbach, H., Bell, B., Berrisford, P., Biavati, G., Horányi, A., Muñoz Sabater, J., Nicolas, J., Peubey, C., Radu, R., Rozum, I., Schepers, D., Simmons, A., Soci, C., Dee, D., Thépaut, J-N., 2018; Fig. S11). It seems that evaporation of small rain drops releases hydrogen peroxide causing elevated hydrogen peroxide mixing ratios at low altitudes.

**5 Conclusions**

A comparison of hydrogen peroxide mixing ratios during the BLUESKY campaign with the previous HOOVER and UTOPIHAN-ACT campaigns shows significant differences within the middle troposphere and the boundary layer. The measurements are only in 30% agreement with previous observations within the lower tropospheric layers. Hydrogen peroxide does not exhibit the expected local maximum at altitudes of 3 – 7 km. The rather constant vertical distribution of the mixing

ratio is most likely related to the enhanced presence of clouds and the subsequent wet scavenging during the measurement period relative to previous air-borne studies.

The measured hydrogen peroxide mixing ratios agree with those simulated by EMAC within the upper troposphere and the boundary layer. The model simulations partly reproduce the strong effect of cloud uptake and rainout loss of the species in the middle troposphere. The calculated deposition loss rates based on EMAC reveal an underestimation relative to the observations



indicating difficulties in the simulation of wet scavenging by the model. This was confirmed by the discrepancies between the rain rates and $H_2O_2$ scavenging values simulated by EMAC and ERA5 meteorological reanalysis data.

While the BLUESKY campaign was performed under lockdown conditions, with substantially reduced anthropogenic emissions, particularly of $NO_x$, we find that reduced $H_2O_2$ mixing ratios compared to the HOOVER and UTOPIHAN-ACT
campaigns are not explained by chemical but rather by meteorological conditions. The importance of rain as a $H_2O_2$ sink, but potentially also in vertically redistributing $H_2O_2$, was shown in a case study based on aircraft measurements over Central Germany. While precipitation scavenging removed $H_2O_2$ from the cloud layer, the evaporation of drizzle droplets in the boundary layer beneath locally increased $H_2O_2$ mixing ratios.


**Data availability.** The data presented in this paper are available from the contact authors under request.

**Author contributions.** JL and HF planned the campaign; ZH, FO and BB performed the measurements; ZH and HF designed the study; ZH, FO, BB, BS analyzed the data; AP developed the model code and performed the simulation; ZH wrote the
manuscript draft with contributions of all co-authors; JL, HF, AP, BB, FO and BS reviewed and edited the manuscript.

**Competing interests.** The authors declare that they have no conflict of interest.

**Acknowledgments.** The authors are very grateful to the BLUESKY team, the Forschungszentrum Jülich, Karlsruhe Institute
of Technology and Deutsches Zentrum für Luft- und Raumfahrt (DLR) in Oberpfaffenhofen for their great support. Their work was essential for the BLUESKY project. Special thanks to Ovid Oktavian Krüger, Department of Multiphase Chemistry, Max Planck Institute for Chemistry for providing the trajectories and the rain rates along the flight tracks.

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
