# Peer review of "Distribution of hydrogen peroxide over Europe during the BLUESKY aircraft campaign"

_Atmospheric Chemistry and Physics, 2022_

## Referee Comment (RC1)

Review of ***Distribution of hydrogen peroxide over Europe during the BLUESKY aircraft campaign*** by Zaneta Hamryszczak et al.

**Significance:**

This manuscript describes airborne measurements of H2O2 and CH3OOH during BLUESKY campaign, and compares the results to common atmospheric model predictions. The study was performed during Covid lockdown in Europe, and thus provides an interesting comparison to research performed previously without lockdown measures.

The manuscript offers an interesting view on the atmospheric peroxide chemistry. The authors seem to be experts in the field of airborne measurements, and the study is performed with previously introduced standard techniques. However, there seems to be some discrepancies with the previous data that could perhaps be caused by instrumental biases, and thus there are several issues I would like the authors to elaborate on, before I can recommend publishing the manuscript. These concerns are detailed below.

**Major comments:**

Hydrogen peroxide and organic peroxide are known to decay on steel surfaces, yet the inlet here is made of steel. How much did this inlet system affect the overall results of this study? Is the 0.52 H2O2 sampling efficiency related to this fact? Does it account the steel part of the sampling? As the inlet system seems critical for understanding the results, it should be better described. A figure would help.

Figure 2: There seems to be very little variation in the obtained values. Is it possible that the instrument was not working correctly? Could you show us the relevant calibration plots, or any other data that shows a time-period where the signal varied considerably?
Additionally, was there a correction term / procedure included to the measurement methodology after the previous flight campaigns, or is the analysis of the peroxides exactly the same between the campaigns? The BLUESKY data seems to consistently report lower H2O2 than other campaigns.

Line 248: Also "intercomparison is dominated by the high variability of the mixing ratios". They are not apparent from the provided figures. Could you explain what you men with this.

It seems from Figure 7a that there was hardly any influence from cloud scavenging or precipitation, as the H2O2 time trace is roughly constant, and both hydroperoxide signals significantly increase during the second cloud "scavenging" episode. This seems to be contradictory to what is discussed. Could you clarify this. Also, It might be that the dimensions of Figure 7a are somewhat too complex (or too reduced?) and reduce its information content. Also, couldn't Fig 7b be interpreted so that the water content actually protects the H2O2 as its concentration steadily increases with altitude all the way to the top of the cloud cover, after which it starts decreasing? This is actually even commented by the authors "Previous studies on the possibility of mass transfer of H2O2 from rain water to the surrounding air indicate a possible release of hydrogen peroxide to the atmosphere (Hua et al., 2008; Huang and Chen, 2010; Xuan et al., 2020).", but the discussion seems a bit misplaced.

**Minor comments:**

Add the instrument used to measure the peroxides already to the abstract. Any other details missing that were crucial for doing the study and/or obtaining the results?

Consider chopping the first paragraph of introduction into several smaller ones.

Line 35: There's an error in describing HOx as "peroxy radicals (HOx),"

"However, the underestimation of the photolysis frequencies by the model can be partly explained by the use of different absorption cross sections of H2O2 (Hottmann et al., 2020)" → Why were different cross- sections used here? Was this explained?

Line 105: Please remove citations to unpublished work (it's not even mentioned in the reference list).
Line 158: What do you mean by "prior to the measurement at 0.95 – 0.98"?
Line 172: Why do you assume ambient H2O2 is zero above tropopause? How valid is this assumption? Could you elaborate.

"Consequently, an increase in the ratio between MHP and hydrogen peroxide of ≥ 1 can ensue as a result of deposition processes within clouds." Is this enough? Why?

Seems a tad bit weird that the peroxide measurements are not mentioned in Table 1.

Line264: Awkward reference. Also in Figure 5 and Line 415, at least.

---

## Referee Comment (RC2)

Review:  Distribution of hydrogen peroxide over Europe during the BLUESKY aircraft campaign, Hamryszczak et al.

The manuscript presents in situ aircraft measurements of hydrogen peroxide and organic peroxides during the BLUESKY campaign in May and June of 2020.  The time period coincides with reduced emissions associated with shutdowns driven by the COVID-19 pandemic.  The authors use a series of models to make the case that cloud scavenging and rainout processes over the region had a greater impact on reducing ambient peroxide levels than a reduction in emissions.  The analytical methods used are sound.  The data set is a valuable contribution and the manuscript is well written, and may be ready for publication with the following clarifications.

Line 137 states the CPI inlet sampling efficiency for hydrogen peroxide was determined to be 0.52.  It would be helpful to know how this was determined and the frequency.  Was the inlet cleaned during the campaign, and did this impact the transmission efficiency?  Was the transmission efficiency examined for organic peroxides?

Line 150, notes an assumption that organic peroxides that pass the inlet are unaffected by any further losses and assumes a stripping efficiency for MHP from Lee et al., 2000.  The manuscript would be strengthened if loss of organic peroxides in the sampling system were characterized.  However short of that details regarding the stripping system should be provided to establish whether adopting the Lee et al stripping efficiency is appropriate.

Line 154, change "…to a lesser extend of…"  to "…extent PAA…"

The analytical method used measures $H_2O_2$ and organic peroxides ($RO_2H$).  The case is made that the $RO_2H$ is likely, largely MHP.  However, the technique does not distinguish the different organic peroxides. I recommend using $RO_2H$ in figure 3 and through out the manuscript.  This will not detract from the arguments presented in the manuscript and will not lead to an impression that MHP was a measured species.

Line 173 notes an instrument interference caused by hopcalite contamination during the campaign.  Can the authors discuss this interference?  Was this interference dependent only on ozone concentrations?  Did this have an impact on the $RO_2H$ channel?

Section 4.3 discusses the fate of peroxides below clouds.  This section could benefit from providing some information and discussion about whether the airmasses sampled above and below the cloud deck have different trajectories and exposure to rain out.  When did drizzle begin in the boundary layer relative to the measurement time?

---

## Author Comment (AC1)

**Please note the used color code**
**(black: RC, red: AC, blue: manuscript changes according to RC recommendations)**

We thank the reviewer for her/his helpful comments.

**Significance:**
This manuscript describes airborne measurements of $H_2O_2$ and $CH_3OOH$ during BLUESKY campaign, and compares the results to common atmospheric model predictions. The study was performed during Covid lockdown in Europe, and thus provides an interesting comparison to research performed previously without lockdown measures.
The manuscript offers an interesting view on the atmospheric peroxide chemistry. The authors seem to be experts in the field of airborne measurements, and the study is performed with previously introduced standard techniques. However, there seems to be some discrepancies with the previous data that could perhaps be caused by instrumental biases, and thus there are several issues I would like the authors to elaborate on, before I can recommend publishing the manuscript.
These concerns are detailed below.

**Major comments:**
Hydrogen peroxide and organic peroxide are known to decay on steel surfaces, yet the inlet here is made of steel. How much did this inlet system affect the overall results of this study? Is the 0.52 $H_2O_2$ sampling efficiency related to this fact? Does it account the steel part of the sampling? As the inlet system seems critical for understanding the results, it should be better described. A figure would help.

The inlet system used on the HALO aircraft is described in detail in the paper by Hottmann et al. (2020): Air was sampled from the top of the aircraft fuselage through a forward facing trace gas inlet (TGI) designed as a bypass, consisting of ½'' PFA (perfluoroalkoxy alkanes) tube inside the aircraft with an exit trough a second TGI. From this bypass ¼'' PFA tube with a flow rate of 2 slpm (standard liter per minute) was directed to HYPHOP. To obtain constant pressure at the HYPHOP inlet a constant pressure inlet (CPI) consisting of a dual stage membrane pump (Vacubrand MD1 VARIO SP, Wertheim, Germany) was used. Similar inlet designs were used for the measurement on the Lear-Jet during UTOPHIAN-ACT and HOOVER (Klippel et al., 2011).
Thus, the inlet system does not include any metal surfaces. Sampling losses are affected mainly by the surface of the ¼" PFA tube and the CPI. We assume that the smaller surface of the bypass which is maintained at a high flow has only a minor influence on $H_2O_2$ inlet losses. Therefore, the inlet efficiency due to losses in the CPI were measured every second day with a gas phase calibration device (for details see our answer to referee 2).

Figure 2: There seems to be very little variation in the obtained values. Is it possible that the instrument was not working correctly? Could you show us the relevant calibration plots, or any other data that shows a time-period where the signal varied considerably?
Additionally, was there a correction term / procedure included to the measurement methodology after the previous flight campaigns, or is the analysis of the peroxides exactly the same between the campaigns? The BLUESKY data seems to consistently report lower $H_2O_2$ than other campaigns.

As discussed in the manuscript, the enhanced presence of clouds within the middle troposphere and relatively high total precipitation rate can affect the levels of hydrogen peroxide and lead to a general decrease in their latitudinal variability relative to that observed in previous campaigns.

Due to the relatively high difference in the range of the means for each campaign dataset, the variability of the $H_2O_2$ values during BLUESKY appears to be lower in comparison to previous campaigns. Fig. S3 of the Supplement gives additional insight into the species variability in each tropospheric layer during the campaign with a higher resolution.

An exemplary signal variation during the measurement flight performed on 23.05.2020 is presented below. Please note that the measured mixing ratios of organic peroxides here were not scaled according to the MHP sampling efficiency of 0.6. In order to avoid data loss, the time consuming liquid calibration was performed prior to the take-off.

[Figure]

Figure 1: Temporal series of BLUESKY flight #1 performed on 23.05.2020. Data were plotted for observed hydrogen peroxide (red) and unscaled sum of organic peroxides (blue) mixing ratios in relation to altitude (top black). Liquid calibration, background measurements and vertical profiles are highlighted in gray, dashed and green boxes, respectively. Please note that the displayed peroxide data has a temporal resolution of 1 sec.

Additional exemplary signal variations during two flights are presented in Fig. 7a of the manuscript as well as in Fig. S10, indicating variations of $H_2O_2$ mixing ratios over almost an order of magnitude (0.1 ppbv to 1 ppbv).

The analysis of the peroxide data was performed analogously to previous campaigns. The measurement methods and instrumentation are based on the same principles.

Line 248: Also "intercomparison is dominated by the high variability of the mixing ratios". They are not apparent from the provided figures. Could you explain what you men with this.

We apologize for the confusion. The mixing ratio means (± 1 sigma) and medians calculated for each campaign subdivided into the main tropospheric layers display a high variability relative to each other, as discussed in detail later in the paragraph. A corresponding overview on estimated means and medians is given in the Table S1 of the Supplement.

Line 254 (former 248) was changed to:

In both lower tropospheric layers (0 – 6 km) the hydrogen peroxide mixing ratios during BLUESKY differed significantly from those measured previously.

It seems from Figure 7a that there was hardly any influence from cloud scavenging or precipitation, as the $H_2O_2$ time trace is roughly constant, and both hydroperoxide signals significantly increase during the second cloud "scavenging" episode. This seems to be contradictory to what is discussed.

Could you clarify this. Also, it might be that the dimensions of Figure 7a are somewhat too complex (or too reduced?) and reduce its information content. Also, couldn't Fig 7b be interpreted so that the water content actually protects the $H_2O_2$ as its concentration steadily increases with altitude all the way to the top of the cloud cover, after which it starts decreasing? This is actually even commented by the authors "Previous studies on the possibility of mass transfer of $H_2O_2$ from rain water to the surrounding air indicate a possible release of hydrogen peroxide to the atmosphere (Hua et al., 2008; Huang and Chen, 2010; Xuan et al., 2020).", but the discussion seems a bit misplaced.

Fig. 7a shows a temporal series of the measurements over Frankfurt, where a vertical profile flight was performed. Please note that high resolution data reported by the instrument are shown. Due to the time resolution of the instrument (90 sec) individual data points are not independent from previous or following data points, leading to slow and small changes, that give the impression of rather constant mixing ratios. During the measurement the aircraft passed the cloud layer during the descending and ascending legs twice. Please note the plotted altitude in the upper part of the figure (black). As mentioned in the introduction of the manuscript the vertical distribution of hydrogen peroxide normally displays a characteristic inverted c-shaped trend. Based on the expected trend, decreasing hydrogen peroxide levels were expected with decreasing flight altitudes, as observed within the measured organic peroxides levels and $H_2O_2$ simulation by EMAC (blue and black plots in the lower part of the figure). Instead, here the measured hydrogen peroxide levels increased at low altitudes, contrary to the expected distribution.

The rise of hydrogen peroxide levels during the second cloud scavenging episode is due to the ascending character of the flight leg and the corresponding increase in altitude and corresponds with the levels of hydrogen peroxide measured during the descending through the cloud layer.

The levels of hydrogen peroxide within the cloud layer are decreasing with the increasing liquid water content towards the base of the cloud. The striking feature here is more likely the rapid increase of hydrogen peroxide concentrations directly below the cloud, which we assume is due to mass transfer from the falling rain droplets to the surrounding air.

Section 4.3. changed to:

The distribution of hydrogen peroxide above, in and below clouds at Frankfurt Airport (50° 1′ 59″ N and 8° 34′ 14″ O) was measured during the BLUESKY-flight #1 and showed untypical increases in hydrogen peroxide mixing ratios at low altitudes.

Based on NOAA HYSPLIT backward trajectory analysis (model duration of 24 h), the probed airmasses originated from the North Atlantic, passing northern France and were nearly uniformly affected by rainout during 6 hours prior to the measurement time. During the measurement the aircraft passed a cloud layer at approximately 2 – 6 km during descending and ascending legs of the vertical profile. The descent and ascent into and out of Frankfurt took place between 9:00 and 11:00 UTC. Fig. 7 displays the time series of the approach to Frankfurt. Mixing ratios of $H_2O_2$ from observations and EMAC are shown.

The relative humidity (RH) of 100% (grey areas in Fig. 7) indicates the presence of clouds. Rain was mainly observed below the clouds at low altitudes (light blue areas) at slightly lower RH. ERA 5 reanalysis (Fig. S11a) confirmed the presence of clouds at altitudes of 2 – 6 km (Flight #1). Based on local meteorological reports, light

rain started approximately one hour prior to the vertical profile measurement and lasted until approximately half an hour.

**Minor comments:**
Add the instrument used to measure the peroxides already to the abstract. Any other details missing that were crucial for doing the study and/or obtaining the results?

The information on the used instrument was added to the abstract. No other details of importance are missing.

Abstract was changed to:
**Abstract.** *In this work we present airborne in situ trace gas observations of hydrogen peroxide ($H_2O_2$), and of the sum of organic hydroperoxides over Europe during the Chemistry of the Atmosphere – Field Experiments in Europe (CAFE-EU, also known as BLUESKY) aircraft campaign using a wet chemical monitoring system, HYdrogen Peroxide and Higher Organic Peroxide monitor (HYPHOP).*
The campaign took place in May/June 2020 over *central* and *southern* Europe with two additional flights dedicated to the North Atlantic Flight Corridor. Airborne measurements were performed on the High Altitude and LOng-range (HALO) research operating out of Oberpfaffenhofen (*southern* Germany). We report average mixing ratios for $H_2O_2$ of $0.32 \pm 0.25$ ppbv, $0.39 \pm 0.23$ ppbv and $0.38 \pm 0.21$ ppbv in the upper and middle troposphere and the boundary layer over Europe, respectively. Vertical profiles of measured $H_2O_2$ reveal a significant decrease in particular above the boundary layer, *contrary* to previous observations, most likely due to cloud scavenging and subsequent rainout of soluble species. In general, the expected inverted c-shaped vertical trend with maximum hydrogen peroxide mixing ratios at $3 – 7$ km was not found during BLUESKY. This *deviates from* observations during previous airborne studies over Europe, i.e., $1.64 \pm 0.83$ ppbv during the HOOVER campaign and $1.67 \pm 0.97$ ppbv during UTOPIHAN-ACT II/III. Simulations with the global chemistry-transport model EMAC partly reproduce the strong effect of rainout loss on the vertical profile of $H_2O_2$. A sensitivity study without $H_2O_2$ scavenging performed using EMAC confirms the strong influence of clouds and precipitation scavenging on hydrogen peroxide concentrations. Differences between model simulations and observations are most likely due to difficulties in the simulation of wet scavenging processes due to the limited model resolution.

Consider chopping the first paragraph of introduction into several smaller ones.

The first paragraph of the introduction was modified as recommended in the RC.

Line 35: There's an error in describing HOx as "peroxy radicals (HOx),"

The error has been corrected.

Line 35 was changed to:
Furthermore, gas-phase hydroperoxides are a reservoir for hydrogen oxide and peroxide radicals ($HO_x$), which are well known for their contribution to the self-cleaning properties of the atmosphere (Levy, 1971; Lelieveld and Crutzen, 1990; Crutzen et al., 1999).

"However, the underestimation of the photolysis frequencies by the model can be partly explained by the use of different absorption cross sections of H2O2 (Hottmann et al., 2020)" à Why were different cross- sections used here? Was this explained?

As reported by Hottmann et al. (2020), the $H_2O_2$ absorption cross sections were extrapolated up to 370 nm based on the recommended wavelength range of 280 – 350 nm in order to capture the whole photolytic activity range of the species.

Line 379 changed to:
However, the underestimation of the photolysis frequencies by the model can be partly explained by the use of additional extrapolated absorption cross sections of $H_2O_2$ in order to reproduce the entire photolytic activity range of the species (Hottmann et al., 2020).

Line 105: Please remove citations to unpublished work (it's not even mentioned in the reference list).

The citation was updated and added to the reference list.

Line 107 (former 105) changed to:
The reduced pollution levels gave rise to anomalous blue skies, hence the name "BLUESKY" (Voigt et al., 2022).

Line 158: What do you mean by "prior to the measurement at 0.95 – 0.98"?

We apologize for the confusing choice of words.

Line 161 (former 158) changed to:
The catalase efficiency for the destruction of $H_2O_2$ in Channel B was determined via liquid calibration of the instrument at 0.95 – 0.98.

Line 172: Why do you assume ambient H2O2 is zero above tropopause? How valid is this assumption? Could you elaborate.

We believe the assumption is justified by general trends in the atmospheric distribution of water vapor and photolytic activity, which are the limiting factors of hydrogen peroxide production at high altitudes. Above the tropopause, the concentration of water vapor decreases drastically due to dehydration processes occurring at the tropopause (Schoeberl and Dessler, 2011; Park et al., 2021). At the same time photolytic activity simultaneously increases, and the role of hydroperoxides as a source of $HO_x$ becomes more prominent, leading to a decrease in hydrogen peroxide levels close to zero.

Line 176 (former 172) was changed to:
The interference was derived by plotting hydrogen peroxide mixing ratios vs. ozone mixing ratios in the lower stratosphere, assuming that ambient $H_2O_2$ is close to zero above the tropopause based on the decreased availability of water vapor for the $H_2O_2$ precursor production and simultaneously increased photolytic activity of $H_2O_2$.

"Consequently, an increase in the ratio between MHP and hydrogen peroxide of $\geq 1$ can ensue as a result of deposition processes within clouds." Is this enough? Why?

The levels of MHP are generally lower than the levels of $H_2O_2$, which leads to a MHP-to-$H_2O_2$-ratio of <1, as displayed in Fig.3c during the HOOVER and UTOPIHAN campaigns. As discussed in Line 272, both campaigns were performed under nearly cloud-free conditions. A significant increase in the ratio, as observed during the BLUESKY campaign, indicates highly decreased levels of $H_2O_2$ relative to MHP. As further discussed in Line 276 MHP is far less sensitive to wet deposition processes than $H_2O_2$ due to a much lower Henry's coefficient. Consequently, as $H_2O_2$ decreases within a cloud layer due to scavenging, MHP remains nearly unaffected. Thus, we believe MHP-to-$H_2O_2$-ratio of ≥ 1 serves as a valid indicator for meteorological changes in terms of clouds and rain in this context.

Seems a tad bit weird that the peroxide measurements are not mentioned in Table 1.

We apologize for the confusion. Table 1 gives a brief overview on significant information of instrumentation providing the supplementary information used in this study.

Table 1 caption was changed to:
Table 1. Overview of other observed species with corresponding measurement method, total measurement uncertainty (TMU) and references regarding the supplementary instrumentation.

Line264: Awkward reference. Also, in Figure 5 and Line 415, at least.

Reference was changed upon the request.

Line 264 (now Line 271), Figure 5 and line 415 (now Line 424) were changed to:
(Hersbach et al. 2018)

**References**

Hottmann, B., Hafermann, S., Tomsche, L., Marno, D., Martinez, M., Harder, H., Pozzer, A., Neumaier, M., Zahn, A., Bohn, B., Stratmann, G., Ziereis, H., Lelieveld, J., and Fischer, H.: Impact of the South Asian monsoon outflow on atmospheric hydroperoxides in the upper troposphere, Atmos. Chem. Phys., 20, 12655–12673, https://doi.org/10.5194/acp-20-12655-2020, 2020.

Park, M., Randel, W. J., Damadeo, R. P., Flittner, D. E., Davis, S. M., Rosenlof, K. H., Livesey, N., Lambert, A., and Read, W.: Near-Global Variability of Stratospheric Water Vapor Observed by SAGE III/ISS, Geophys Res Atmos, 126, https://doi.org/10.1029/2020JD034274, 2021.

Schoeberl, M. R. and Dessler, A. E.: Dehydration of the stratosphere, Atmos. Chem. Phys., 11, 8433–8446, https://doi.org/10.5194/acp-11-8433-2011, 2011.

Voigt, C., Lelieveld, J., Schlager, H., Schneider, J., Curtius, J., Meerkötter, R., Sauer, D., Bugliaro, L., Bohn, B., Crowley, J. N., Erbertseder, T., Groβ, S., Hahn, V., Li, Q., Mertens, M., Pöhlker, M. L., Pozzer, A., Schumann, U., Tomsche, L., Williams, J., Zahn, A., Andreae, M., Borrmann, S., Bräuer, T., Dörich, R., Dörnbrack, A., Edtbauer, A., Ernle, L., Fischer, H., Giez, A., Granzin, M., Grewe, V., Harder, H., Heinritzi,

M., Holanda, B. A., Jöckel, P., Kaiser, K., Krüger, O. O., Lucke, J., Marsing, A., Martin, A., Matthes, S., Pöhlker, C., Pöschl, U., Reifenberg, S., Ringsdorf, A., Scheibe, M., Tadic, I., Zauner-Wieczorek, M., Henke, R., and Rapp, M.: Cleaner skies during the COVID-19 lockdown, Bulletin of the American Meteorological Society, https://doi.org/10.1175/BAMS-D-21-0012.1, 2022.

---

## Author Comment (AC2)

**Please note the used color code**
**(black: RC, red: AC, blue: manuscript changes according to RC recommendations)**

We thank the reviewer for her/his helpful comments.

The manuscript presents in situ aircraft measurements of hydrogen peroxide and organic peroxides during the BLUESKY campaign in May and June of 2020. The time period coincides with reduced emissions associated with shutdowns driven by the COVID-19 pandemic. The authors use a series of models to make the case that cloud scavenging and rainout processes over the region had a greater impact on reducing ambient peroxide levels than a reduction in emissions. The analytical methods used are sound. The data set is a valuable contribution and the manuscript is well written, and may be ready for publication with the following clarifications.

Line 137 states the CPI inlet sampling efficiency for hydrogen peroxide was determined to be 0.52. It would be helpful to know how this was determined and the frequency. Was the inlet cleaned during the campaign, and did this impact the transmission efficiency? Was the transmission efficiency examined for organic peroxides?

As mentioned in the manuscript (Line 164), the sampling efficiency was determined using gas phase calibration source, consisting of a LDPE permeation device filled with 30% hydrogen peroxide in a temperature-controlled oven at 35 °C flushed with synthetic air at a rate of 60 standard cubic centimeters per minute which was further diluted with approximately 2300 sscm purified air. The sampling efficiency was measured every second day during the field campaign by attaching the permeation source in front and behind the CPI inlet and calculated from the difference between the measured hydrogen peroxide levels with and without the CPI inlet implemented to the gas flow. Please note that the CPI and the ¼" PFA tubing connecting it to the ½" bypass line (also PFA) represent the largest surface of the whole inlet design, so that we assume that surface losses dominantly occur in this part of the inlet, while the high flow in the bypass minimizes surface losses there. The whole inlet was regularly flushed with purified dry air at rates of approximately 10,000 sscm. No impact on the transmission efficiency was observed based on single measurements. Due to lack of adequate organic peroxide permeations sources during the campaign the transmission efficiency of organic peroxides was not examined.

Line 164 (former 160) changed to:
In order to estimate the sampling efficiency, a calibration gas was analyzed every second day during the field campaign. The calibration gas was created by a LDPE permeation source filled with 30% hydrogen peroxide embedded in a temperature-controlled oven at 35 °C and flushed with synthetic air at a rate of 60 standard cubic centimeters per minute (sccm). The defined amount of hydrogen peroxide gas was diluted with approximately 2300 sccm purified ambient air. The sampling efficiency was calculated based on the difference between the measured hydrogen peroxide levels with and without the CPI inlet implemented into the calibration gas flow. The permeation gas is calibrated by bubbling the gas through a water-filled flask followed by photometric examination via UV spectroscopy using the $TiCl_4$ method described by Pilz and Johann (1974).

Line 150, notes an assumption that organic peroxides that pass the inlet are unaffected by any further losses and assumes a stripping efficiency for MHP from Lee et al., 2000. The manuscript would be strengthened if loss of

organic peroxides in the sampling system were characterized. However short of that details regarding the stripping system should be provided to establish whether adopting the Lee et al stripping efficiency is appropriate.

As mentioned above, gas phase calibration devices for organic peroxides were not available. We expect that individual ROOH will have a different sampling efficiency depending on wall losses in the inlet and solubility into the scrubbing solution. Assuming that MHP is the dominant component of ROOH, we scale the scrubbing efficiency relative to $H_2O_2$ by using the Henry's law constant of those species. The sampling set-up, with respect to the sampling coil, the used sampling solution, the residence time and the temperature are similar to the design used in Lee et al. (2000) (gas flow rate of approximately 2300 sccm, scrubbing glass coil continuously flushed with precooled buffered sampling solution consisting of 41 g potassium hydrogen phthalate, 185 mL 1 M NaOH, 0.1 g EDTA and 1 mL of 37% HCHO at a flow rate of 0.000508 L/min (pH 6) (Lazrus et al., 1986)). Therefore, we assume a similar sampling efficiency as used in Lee et al. (2000).

[revised manuscript text omitted]

Figure S10 changed to:

[Figure]

BORD #8

Caption of Figure S10 changed to:

**Figure S10: Temporal series of peroxide mixing ratios over the Bordeaux area on 09.06.2020 (red: observed $H_2O_2$; blue: observed ROOH; black: simulated $H_2O_2$, grey: relative humidity; top black: GPS altitude). Cloud scavenging and precipitation are highlighted by gray and light blue shading, respectively. Please note that the observed peroxide data displayed has 1 sec time resolution in comparison to the model resolution of 5 min.**

Line 173 notes an instrument interference caused by hopcalite contamination during the campaign. Can the authors discuss this interference? Was this interference dependent only on ozone concentrations? Did this have an impact on the $RO_2H$ channel?

Hopcalite contains of a variety of metal oxides used to purify the ambient air from the majority of trace gases during background measurements. Since peroxides are sensitive to metal oxides, a lowered signals and therefore decreased levels of the species due to the contamination were observed. To our knowledge, the interference is not dependent on the ozone levels, but rather on the amount of metal oxides causing the contamination. Since the amount of the released hopcalite was not quantified, the impact of the negative interference could not be estimated separately and the ozone interference was extended by the hopcalite interference as a rough estimate relative to the calculated total measurement uncertainty.

Line 179 (former173) changed to:
Due to instrumental issues caused by hopcalite contaminations during the campaign, the uncertainty of the ozone interference was further extended by the hopcalite interference and estimated as 27% at 0.16 ppbv hydrogen peroxide.

Section 4.3 discusses the fate of peroxides below clouds. This section could benefit from providing some information and discussion about whether the airmasses sampled above and below the cloud deck have different trajectories and exposure to rain out. When did drizzle begin in the boundary layer relative to the measurement time?

Based on NOAA HYSPLIT (Stein et al., 2015) backward trajectories and GFSQ meteorological data analysis above, in and below the cloud layer (500 m, 300 m and 8000 m) with a duration of 24 h, the majority of the airmasses originated from North Atlantic passing North France. The analysis confirmed further a comparable exposure to rainfall in all layers. The drizzle begin was reported approximately 1 hour earlier and lasted until 11:00 UTC where the precipitation rate increased for another 2 hours.

Section 4.3. changed to:

The distribution of hydrogen peroxide above, in and below clouds at Frankfurt Airport (50° 1′ 59″ N and 8° 34′ 14″ O) was measured during the BLUESKY-flight #1 and showed untypical increases in hydrogen peroxide mixing ratios at low altitudes.

Based on NOAA HYSPLIT backward trajectory analysis (model duration of 24 h), the probed airmasses originated from the North Atlantic passing northern France and were nearly uniformly affected by rainout during 6 hours prior to the measurement. During the measurement the aircraft passed a cloud layer at approximately 2 – 6 km during descending and ascending lags of the vertical profile. The descent and ascent into and out of Frankfurt took place between 9:00 and 11:00 UTC. Fig. 7 displays the time series of the approach to Frankfurt. Mixing ratios of $H_2O_2$ from observations and EMAC are shown.

The relative humidity (RH) of 100% (grey areas in Fig. 7) indicates the presence of clouds. Rain was mainly observed below the clouds at low altitudes (light blue areas) at slightly lower RH. ERA 5 reanalysis (Fig. S11a) confirmed the presence of clouds at altitudes of 2 – 6 km (Flight #1). Based on local meteorological reports, light rain started approximately one hour prior to the vertical profile measurement and lasted until approximately half an hour.